# Polarization Insensitive, Broadband, Near Diffraction-Limited Metalens in Ultraviolet Region

**DOI:** 10.3390/nano10081439

**Published:** 2020-07-23

**Authors:** Saima Kanwal, Jing Wen, Binbin Yu, Xu Chen, Dileep Kumar, Yi Kang, Chunyan Bai, Saima Ubaid, Dawei Zhang

**Affiliations:** 1Engineering Research Center of Optical Instrument and Systems, Ministry of Education and Shanghai Key Lab of Modern Optical System, University of Shanghai for Science and Technology, No. 516 Jun Gong Road, Shanghai 200093, China; 142499021@st.usst.edu.cn (S.K.); jwen@usst.edu.cn (J.W.); 161390013@st.usst.edu.cn (B.Y.); 191380026@st.usst.edu.cn (X.C.); 171390035@st.usst.edu.cn (Y.K.); 171390036@st.usst.edu.cn (C.B.); 161399008@st.usst.edu.cn (S.U.); 2State Key Laboratory of Industrial Control Technology, College of Control Science and Engineering, Zhejiang University, Hangzhou 310027, China; 11332042@zju.edu.cn; 3Shanghai Institute of Intelligent Science and Technology, Tongji University, Shanghai 200093, China

**Keywords:** UV, metalens, polarization insensitive, broadband, dielectric, diffraction-limited

## Abstract

Metasurfaces in the ultraviolet spectrum have stirred up prevalent research interest due to the increasing demand for ultra-compact and wearable UV optical systems. The limitations of conventional plasmonic metasurfaces operating in transmission mode can be overcome by using a suitable dielectric material. A metalens holds promising wavefront engineering for various applications. Metalenses have developed a breakthrough technology in the advancement of integrated and miniaturized optical devices. However, metalenses utilizing the Pancharatnam–Berry (PB) phase or resonance tuning methodology are restricted to polarization dependence and for various applications, polarization-insensitive metalenses are highly desirable. We propose the design of a high-efficiency dielectric polarization-insensitive UV metalens utilizing cylindrical nanopillars with strong focusing ability, providing full phase delay in a broadband range of Ultraviolet light (270–380 nm). The designed metalens comprises Silicon nitride cylindrical nanopillars with spatially varying radii and offers outstanding polarization-insensitive operation in the broadband UV spectrum. It will significantly promote and boost the integration and miniaturization of the UV photonic devices by overcoming the use of Plasmonics structures that are vulnerable to the absorption and ohmic losses of the metals. The focusing efficiency of the designed metalens is as high as 40%.

## 1. Introduction

Ultraviolet light plays a vital part in human life involving sterilizers, lithography equipment, and laser devices. Lenses are one of the essential components among the UV equipment for convergence of UV light. Traditional optical components and lenses use the refractive optical scheme i.e., gradual phase accumulation for the wavefront shaping during the propagation of the light which results in expensive, bulky, and low-efficiency lenses [1]. On the other hand, metasurfaces that are ultra-thin optical structures have appeared to provide a substantial way, by discrete phase changes over the subwavelength scale, to control the amplitude, phase and polarization of the light [2,3,4] to realize optical functionalities by their planar structure [5] such as meta-holograms [6], axicons metalenses [7], antireflection coatings color filters [8], color imaging [9,10], optical vortex generators and polarizers [11,12] and numerous novel photonic devices and systems [13]. Metasurfaces have gained substantial interest in the past few years [14,15] in having the ability to replace the bulky, expensive, low-efficiency optical components [7,16] and provide a gateway towards the miniaturization of the optical devices and components [17,18]. Moreover, metasurfaces are greatly desired in integrated optics and electronic circuits, portals, and mobile devices. They are attaining a great interest in industrial applications due to their high efficiency of manipulating the light [19,20].

Electromagnetic phase control is the most distinctive and perhaps the simplest application of the metasurfaces [21]. The key to implementing various applications is the control of the full 2*π* phase. There are numerous ways to modulate the phase of the electromagnetic waves in a convenient manner, such as geometrical phase control and dynamical phase control. Geometrical phase metasurfaces that are proposed and experimentally demonstrated, utilize the employment of a dominant electrical resonance by asymmetric nanopillars to induce local phase retardation by rotating the polarization of incident light. This sort of phase retardation, that is also called the Pancharatnam–Berry (PB) phase shift, can be employed for beam shaping and spin filter applications. Employing this method, efficiency as high as 86% is validated experimentally for lensing. Transmissive metasurfaces designed by the PB phase method have been demonstrated in IR and visible range with anomalous properties, but this method is specifically restricted to circularly polarized incident light. The second approach that is used to modulate the phase of light is dynamical phase control, which is independent of the polarization of the incident light. This approach attains the desired optical path difference by changing the optical path of the light [22]. Interfacial phase discontinuities are introduced by the metasurfaces along the optical path as an alternative approach to realize the flat and compact dynamical phase elements

Plasmonic metasurfaces and lenses have a very low efficiency equated to the dielectric metasurfaces due to the high inherent Joule losses at optical frequencies, particularly in transmissive mode [23,24,25]. The energy efficiency of these systems is no higher than 20%. Besides, metals are not stable at higher temperatures, and so they are not an appropriate option for high power laser applications. However, these losses can be reduced by adopting a dielectric material having a high refractive index, a larger bandgap, and lower absorption losses at the optical frequencies [26].

Depending on the optical properties of the material, previous work on metalenses has focused on diverse wavelengths ranging from Ultraviolet to near-infrared [27]. However, due to insufficient phase delay, these lenses defocused at a short UV wavelength range [28,29,30,31,32], and these do not operate in the broadband UV spectrum. Moreover, they do not support the polarization independent operation of the reported metalens [28,33,34].

We have recently demonstrated a Pancharatnam–Berry phase-based high-efficiency broadband metalens in the UV spectrum (250–400 nm), which requires circularly polarized (CP) incident light [35]. In this article, we demonstrate the design of a polarization insensitive, transmissive planar metasurface lens composed of silicon nitride Si_3_N_4_ cylindrical nanopillars operating at the broadband UV range (270–380 nm). By varying the radius of the nanopillar abrupt phase, changes can be realized. A flat metasurface lens operating in broadband UV spectrum with corresponding focusing efficiency as high as 40% is demonstrated. Our designed dielectric metalens offers brilliant polarization insensitive operation over the entire defined UV spectrum because of the higher symmetry of the unit cells (and so the designed metalens). Moreover, the calculated full width at half maximum (FWHMs) of the focal spot is also near the diffraction limit. Our designed metalens will pave a new way towards the advancement of integration and miniaturization of the UV photonic systems and its applications. To the authors’ knowledge, this is the first reported metalens in the UV regime with polarization independence and broadband (over 110 nm bandwidth) ability along with a high Numerical Aperture (NA) of 0.86.

## 2. Materials and Methods

The optical properties of the selected material are very critical towards the development of a high-efficiency metasurface as specified by complex refractive index *ñ* = *n* + *ik.* The selected material should possess negligible absorption loss (*k* = 0) and a high refractive index (*n* > 2). Despite a negligible absorption, which is critical for high transmission efficiency, a higher refractive index promises strong confinement of the operating UV regime eventually providing full 0–2*π* phase control. Traditional dielectric materials possess a narrow bandgap triggering high absorption losses in UV. Titanium oxide and gallium nitride have fairly large bandgaps, but they are not suitable to use due to their higher absorption losses at the targeted UV regime. Si_3_N_4_ is chosen as a dielectric material for the UV metalens to realize high efficiency in UV spectral region. Due to its wide bandgap of about 5.1 eV, its large bandgap exhibits the wide transparency window by having exceptionally low extinction coefficient (*k* = 0), and a larger refractive index of about 2.5. The optical constants (n, k) of Si_3_N_4_ [36], which can be considered as the fingerprint of the material, are shown in Appendix A.

There exist several options for considering the dielectric nanopillars for the design of metalenses, such as a slit, square cylinder, cuboid, cylinder, combined meta-molecules, and elliptic cylinders. These nanopillars could be divided into two kinds considering their sensitivity of polarization. The ones depending on the polarization of the light have rotation angles and are spatial anisotropic based on the Pancharatnam–Berry phase theory. Whereas the polarization insensitive are always square cylinders or cylinders utilizing side lengths or diameters to map the required phase. In this article, we use cylindrical nanopillars to control the phase of the light. The building blocks of the designed metalens are Si_3_N_4_ nanopillars on a glass substrate shown in Figure 1a–c.

By optimizing the geometric parameters of the nanopillar full 2π phase coverage can be realized. The unit cell dimension (P), nanopillar radius (R), and height (H) are the three basic geometric and structural design parameters that control the phase and the transmission amplitude of the metasurface lens to ensure high efficiency. The design wavelength of the metalens is 290 nm and it is fully operational in the broadband spectrum of the UV light, i.e., 270–380 nm with excellent polarization insensitive behavior. Performing the simulations through commercial finite-difference-time-domain (FDTD) method implemented by commercial software ‘FDTD Solutions’ (produced by Lumerical Solutions Co. Ltd., Vancouver, BC, Canada) [37] the transmission and phase modulation of the nanopillar with different geometric parameters are calculated [35]. The nanopillar must have sufficient height to provide a full 2*π* phase coverage over a range of radii. Besides, the unit cell dimension must be optimized wisely and the radius should be tuned in a manner to acquire the required phase profile. A set of simulations were performed to get the optimized values of the unit cell dimension, the height, and the range of the radius of nanopillar. Periodic boundary conditions were applied in *x* and *y* directions while perfectly matched layer (PML) boundary conditions were applied to the z-direction for calculating the aforesaid unit cell parameters.

## 3. Design, Results, and Discussion

The building blocks of the metalens are Si_3_N_4_ nanopillars on the glass substrate. The metalens operates in transmission mode and it focuses the collimated light into a spot. To realize this, the required phase must be imparted by each nanopillar at the position *(x*, *y*) as given by [38]
(1)$$\phi (x,y)=2\pi -\frac{2\pi }{\lambda_{d}}(\sqrt{x^{2}+y^{2}+f^{2}}-f)$$
where *λ_d_* is the design wavelength and *f* is the focal length. By varying the radius of nanopillars as a function of their position (*x*, *y*) the effective index of the propagating mode is changed to achieve the desired phase profile *ϕ(x, y)* (Equation (1)). To achieve high transmission through the entire defined spectral range other parameters such as unit cell size U and the nanopillar height H are optimized at design wavelength *λ_d_*. The optimized height of the nanopillar must be sufficiently tall so as to realize full 2π phase coverage over a range of radius.

The optimized height of the nanopillar is 600 nm, the radius of the Si_3_N_4_ nanopillar varies between 30 and 92 nm besides, and the unit cell dimension is 200 nm. The transmission and phase response calculated through the simulations at design wavelength *λ* = 290 nm, H = 600 nm, and P = 200 nm are shown in Figure 2a,b, respectively. The transmission and phase response in the broadband region at R = 36 nm can be seen in Figure 2c,d, respectively.

It can be seen via the Figure 2 that the nanopillars are highly transmissive and can acquire the 2*π* phase accumulation. To design the metalens, perfectly matched layer (PML) boundary conditions were applied in the x, y, and z directions. The required phase mask *ϕ*(*x*, *y*) was discretized by presuming square lattice unit cells with dimensions U × U so that the transmission error $\left|T_{m}e^{i\varphi_{t}(xi,yi)}-T(D)e^{i\varphi (D)}\right|$ [38] is minimized where *T_m_* is the averaged transmission. Figure 3 presents the focusing characteristics of the designed metalens. The NA of the metalens at the design wavelength *λ_d_* = 290 nm is calculated to be 0.86. The intensity profiles of the beam in the *x*-*z* cross-section and *x*-*y* cross-section of the metalens are shown in Figure 3.

The intensity distribution of the focal spot shows a tightly focused, symmetric, and bright spot at the center of the focal plane at different wavelengths (270 nm, 298 nm, 334 nm, and 380 nm). Due to the rotational symmetry of the designed metalens, focus spots for the TM and TE polarizations are just the same. Therefore, all the results are only given for TE polarization while omitting TM results.

The focal length of the metalens is calculated at representative wavelengths. At the focal points of the respective wavelengths i.e., 270 nm, 298 nm, 334 nm, and 380 nm, the FWHMs are calculated as 204 nm, 222 nm, 250 nm, and 284 nm. All the values are near diffraction-limited i.e., $\frac{\lambda_{d}}{2\mathrm{NA}}$. The normalized intensity profiles of the focal spot in the *x*-direction (at *z* = f, and *y* = 0) are shown in Figure 4.

The focusing efficiency, which is defined as the ratio of the total light intensity at the focal spot to the transmitted light [8], is as high as 40%. The focusing efficiency of the designed metalens is shown in Figure 5.

In the end, we compared our results with other references in terms of material platform, bandwidth, incident polarization, and NA; the comparison is shown in Table 1. We noticed that some of these previous studies are not polarization insensitive [33,34] while others are not broadband [29,32], and compared to these studies the NA of our designed metalens is the highest as well.

## 4. Conclusions

This work reports on a high efficiency, polarization insensitive, near diffraction-limited UV metalens that operates in the broadband region of ultraviolet light from 270 to 380 nm with a tightly focused spot. The designed metalens is composed of silicon nitride cylindrical nanopillars arranged on a square lattice. The limitations of the plasmonics metalenses, such as Ohmic losses, are overcome by using dielectric silicon nitride nanopillars as the basic building blocks of the metalens. By altering the geometric parameters of the unit cell, high transmission, and sufficient phase coverage is achieved in the target wavelength range. The focusing efficiency is calculated to be as high as 40%. The designed high-efficiency UV metalens can be used in lightweight, low cost, and planar UV devices, such as UV router, lithography, image sensor, UV laser, communication and sterilization, and so on. Achieving a high-efficiency polarization insensitive metalens operating in the broadband UV range is a critical step toward the implementation of the metasurfaces for practical applications in nanophotonics and integrated optics in the UV. This should promote the high-density integration and miniaturization of the UV nanophotonics. In the future, we intend to maximize the focusing efficiency and design an achromatic, broadband UV metalens to incorporate them into daily life, for widespread applications.

## Figures and Tables

**Figure 1 nanomaterials-10-01439-f001:**
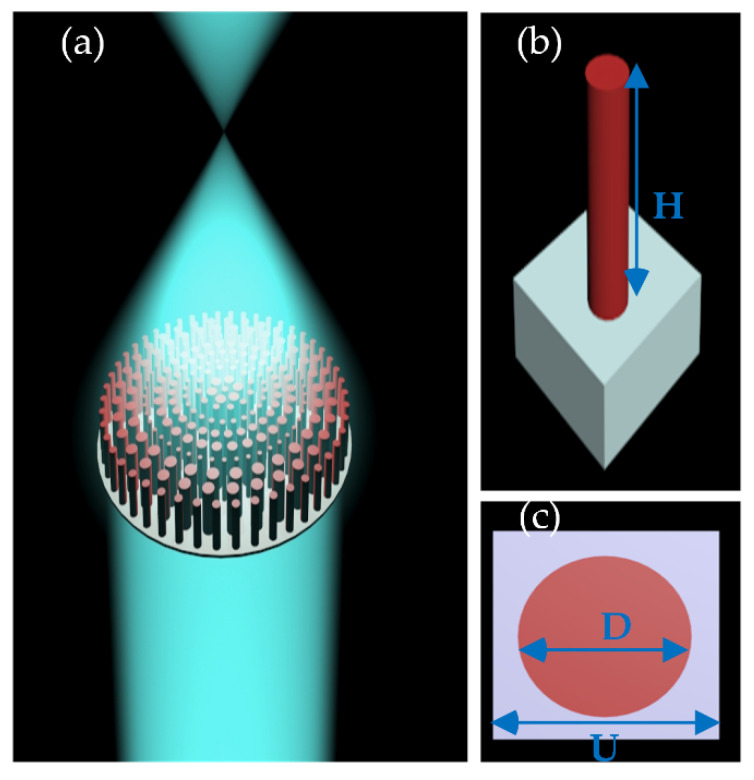
(**a**) Schematic view of the transmissive metalens. (**b**) Side view of the metalens building block. (**c**) Top of the metalens building block. For the designed wavelength of the metalens λ = 290 nm, the nanopillar height H = 600 nm, unit cell dimension P = 200 nm, and the nanopillar radius varies between 30 and 92 nm.

**Figure 2 nanomaterials-10-01439-f002:**
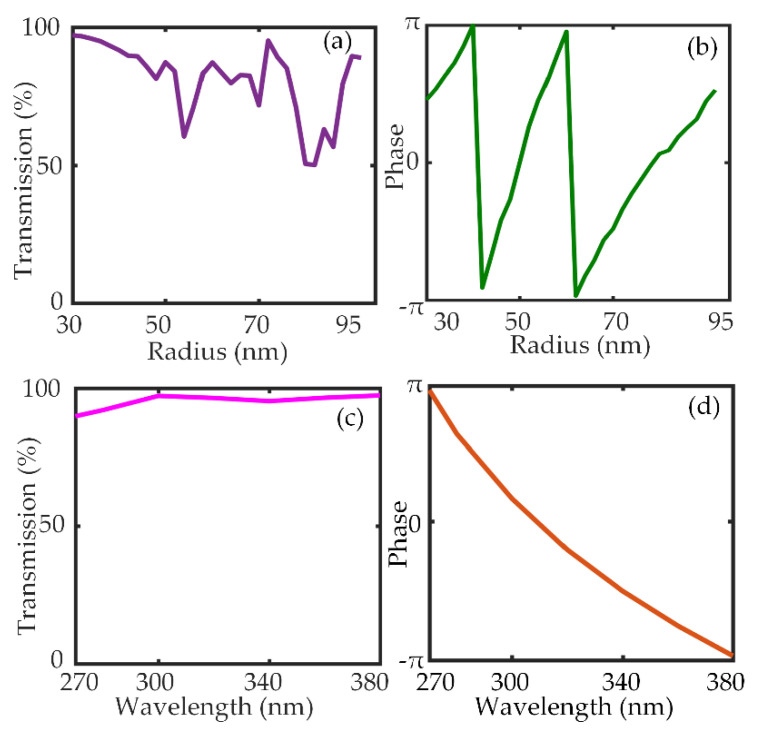
(**a**) Transmission as a function of radius. (**b**) Phase as a function of radius. (**c**) Transmission as a function of wavelength at R = 36 nm. (**d**) Phase as a function of wavelength at R = 36 nm.

**Figure 3 nanomaterials-10-01439-f003:**
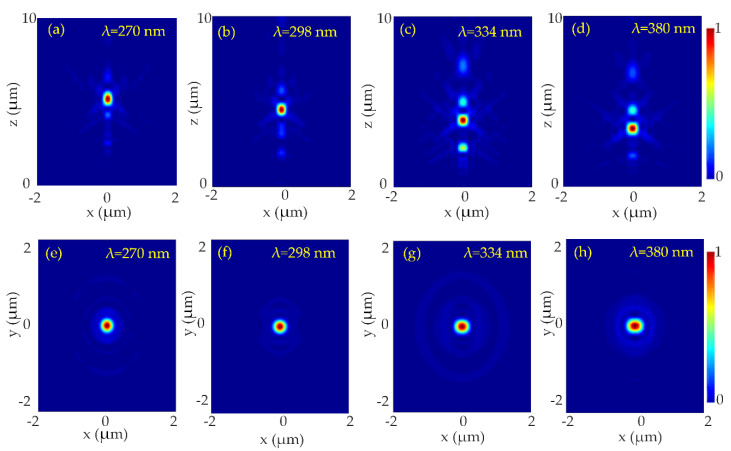
Normalized intensity distribution of the designed UV metalens at incident UV light 270–380 nm at the *x*-*z* plane. (**a**) *λ* = 270 nm, (**b**) *λ* = 298 nm, (**c**) *λ* = 334 nm (**d**) *λ* = 380 nm. The NA of the metalens at designed wavelength *λ_d_* = 290 nm is 0.86. Normalized intensity distribution of the designed UV metalens at incident UV light 270–380 nm at the x-y plane at *x* = *y* = 0. (**e**) *λ* = 270 nm, (**f**) *λ* = 298 nm, (**g**) *λ* = 334 nm (**h**) *λ* = 380 nm. The NA of the metalens at designed wavelength *λ_d_* = 290 nm is 0.86.

**Figure 4 nanomaterials-10-01439-f004:**
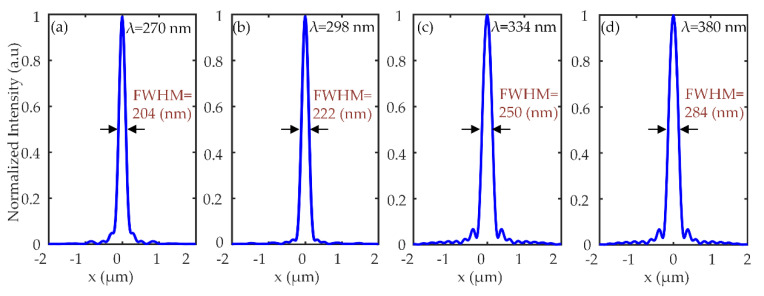
Normalized intensity profile of the focus spot along x-direction at respective wavelengths i.e., 270 nm, 298 nm, 334 nm and 380 nm. (**a**) y = 0, z = 5.3 µm, (**b**) y = 0, z = 4.6 µm, (**c**) y = 0, z = 4 µm and (**d**) y = 0, z = 3.1 µm.

**Figure 5 nanomaterials-10-01439-f005:**
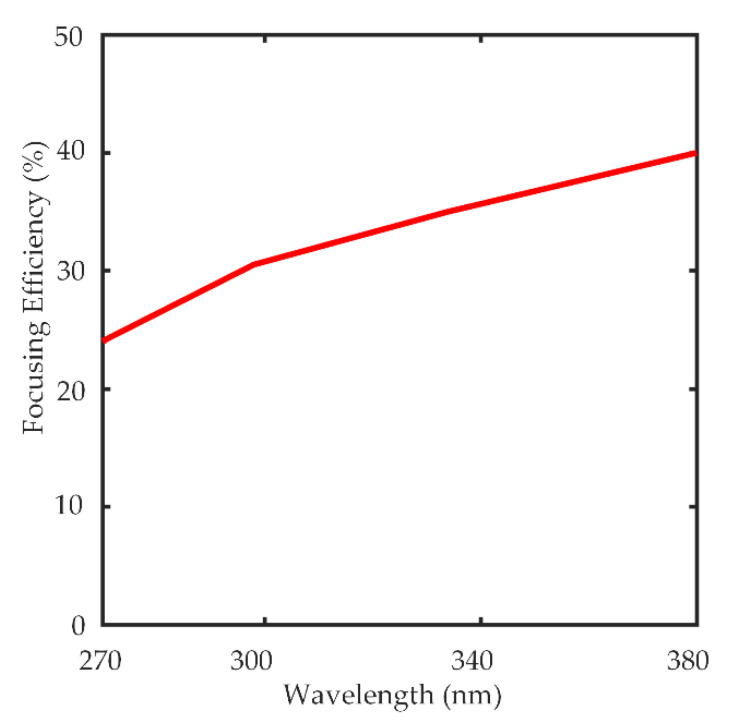
Focusing Efficiency of the metalens designed for the broadband UV spectrum 270–380 nm.

**Table 1 nanomaterials-10-01439-t001:** Summary of our results and other references.

Reference	Material	Wavelength (nm)	Incident Light	NA
Guo et al. [33]	AlN	244–375	CP	>0.1
Huang et al. [34]	Nb_2_O_5_	355	CP	Not mentioned
Hu et al. [31]	AlN	234–274	Polarization Insensitive	>0.1
Zhang et al. [29]	HfO_2_	325	Polarization Insensitive	0.6
Our Work	Si_3_N_4_	270–380	Polarization Insensitive	0.86

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
