# Peer review of "Polarization Insensitive, Broadband, Near Diffraction-Limited Metalens in Ultraviolet Region"

_nanomaterials, 2020, doi:10.3390/nano10081439_

Round 1

Reviewer 1 Report

The authors of paper titled “High-Efficiency, High Numerical Aperture, Near Diffraction-Limited, Broadband, Polarization Insensitive Metalens in Ultraviolet Spectrum” reports a metasurface design for ultraviolet wavelengths and numerically demonstrates a focusing efficiency of 40 %. Even though metasurfaces design for ultraviolet wavelengths is important, the proposed design lacks novelty and obtained results are not better compared to existing designs.    

  1. This work lacks novelty. Authors can find lots of papers, even experimental works [https://doi.org/10.1002/lpor.201800289, https://www.osapublishing.org/abstract.cfm?uri=CLEO_QELS-2019-FM3C.3] in the same direction with much higher performance compared to the present work. In particular, this work is just an extension of the design experimentally reported in Ref [Low loss metasurface optics down to the deep ultraviolet region].
  2. I found that same sentence is repeating several places in the text. In page 3, line 120, A set of simulations was performed through FDTD solutions. Again, in page 4, line 133, The simulations for the metalens were performed by FDTD and line 137, The transmission and phase response calculated through FDTD simulations.    
  3. Authors mentioned “Periodic boundary conditions were applied in x and y directions while PML boundary conditions were applied to the z-direction”. Then mentioned “Perfectly matched layer (PML) boundary conditions were applied in the x, y, and z directions”. Which one is correct?

4.     Lastly, the title of the paper is too long. Try to make it short

Author Response

On behalf of my co-authors, I would like to thank you for giving us the opportunity to revise our manuscript and improve its quality. We have reviewed the comments carefully and revised the manuscript entitled “Polarization Insensitive, Broadband, Near Diffraction-Limited Metalens in Ultraviolet Region," submitted for consideration for publication in ‘nanomaterials’ with manuscript ID: nanomaterials-861452.

Below is our response to the reviewer’s comments and suggestions.

  1. This work lacks novelty. Authors can find lots of papers, even experimental works [https://doi.org/10.1002/lpor.201800289, https://www.osapublishing.org/abstract.cfm?uri=CLEO_QELS-2019-FM3C.3] in the same direction with much higher performance compared to the present work. In particular, this work is just an extension of the design experimentally reported in Ref [Low loss metasurface optics down to the deep ultraviolet region].

Response: According to the reviewer’s suggestion, the manuscript has been updated describing the novelty of our work and its comparison with other studies. Moreover, about the references given by the reviewer as [https://doi.org/10.1002/lpor.201800289, https://www.osapublishing.org/abstract.cfm?uri=CLEO_QELS-2019-FM3C.3], the first one is not polarization insensitive as well as not broadband (only works at 355 nm) while the latter is not broadband (only works at 325 nm). Besides, the [Low loss metasurface optics down to the deep ultraviolet region] is also not a broadband metalens design, it only works at 325 nm with a NA of 0.6.

The manuscript is updated as (Line 88-90) “To the authors' knowledge, this is the first reported metalens in the UV regime with polarization independence and broadband (over 110 nm bandwidth) ability along with a high Numerical Aperture (NA) of 0.86 and, near diffraction-limited focusing characteristics.”

(Line 181-185) “In the end, we compare our results with other references in terms of material platform, bandwidth, incident polarization, and NA; the comparison is shown in Table 1. We notice that some of these previous studies are not polarization-sensitive[33,34] while others are not broadband[29,32], and compared to these studies the NA of our designed metalens is the highest as well.”

Table 1. Summary of our results and other references.

Reference

Material

Wavelength (nm)

Incident Light

NA

Guo et al. [33]

AlN

244-375

CP

> 0.1

Huang et. al [34]

Nb2O5

355

CP

Not mentioned

Hu et al. [31]

AlN

234-274

Polarization Insensitive

> 0.1

Zhang et. al [29]

HfO2

325

Polarization Insensitive

0.6

Our Work

Si3N4

270-380

Polarization Insensitive

0.86

  1. I found that same sentence is repeating several places in the text. In page 3, line 120, A set of simulations was performed through FDTD solutions. Again, in page 4, line 133, The simulations for the metalens were performed by FDTD and line 137, The transmission and phase response calculated through FDTD simulations.

Response: According to the reviewer's suggestion, the repetition of the words are reviewed and revised. (Line 126, Line 141 and, Line 144).

  1. Authors mentioned “Periodic boundary conditions were applied in x and y directions while PML boundary conditions were applied to the z-direction”. Then mentioned “Perfectly matched layer (PML) boundary conditions were applied in the x, y, and z directions”. Which one is correct?

Response: Sorry for the ambiguity, but both of the statements are true as periodic boundary conditions were applied in x and y directions and PML boundary conditions were applied to the z-direction for the simulations of the unit cell parameters optimization (i.e. the unit cell dimension, the height, and the radius of the nanopillar). While PML boundary conditions were applied in all x, y, and, z directions  in the simulations of the metalens.” The manuscript is updated (Line 127-129 and Line 151-152), making it more clear.

  1. Lastly, the title of the paper is too long. Try to make it short.

Response: As suggested by the reviewer to shorten the title, it is modified as “Polarization Insensitive, Broadband, Near Diffraction-Limited Metalens in Ultraviolet Region”.

Reviewer 2 Report

“High efficiency, high N. A., near-diffraction limited, broadband, polarization insensitive metalens in UV spectrum”

By S. Kanwal et al.

Referee Report

The manuscript describes a very interesting study about a subject of high topical importance, namely the construction of optical metasurfaces, i. e. nanostructured planar structures which can replace versatile optical elements and have a high capability of miniaturization and integration into more complex optical systems. The study focusses on a dielectric metasurface (thereby circumventing losses that are known to appear in plasmonic structures), which is claimed to be particularly suitable for applications in the ultraviolet region. This is a very important topic.

However, the manuscript needs some mandatory improvements before it can be considered for publication.

  • It is not clear, whether this manuscript describes a theoretical or an experimental work. Certainly, it makes a difference, whether some hypothetical properties are calculated or whether the theoretical expectations are matched by experimental proof. Some phrases, like “we propose …”, “the designed metalens …”, “… are calculated” seem to indicate that only results of computer simulations are presented, while other phrases, like “… were realized”, “To build the metalens …”, or “S. K. … designed the experiments” seem to indicate that experiments have been performed. This is very confusing. Please indicate either in general or for each specific results, whether it is obtained from calculation or from an experiment.
  • If an experiment was performed, please, describe the experimental setup.
  • The results obtained by calculations depend essentially on the material properties that were assumed. Please, indicate the latter explicitly (at least in a supplementary information), rather than just indicating “the refractive index of Si3N4 is obtained from [30].
  • It is claimed that an optimum design was obtained. However, this design is not given. Please, indicate explicitly the diameter of the nanopillars as a function of radial distance from the center of the metalens, for which the high efficiency was obtained.
  • Reference [31] by the same authors and with a very similar title was published in Nanomaterials, recently. Please, indicate the difference and the progress of the current manuscript in comparison with this previous work.
  • The 3rd sentence in section 3 (lines 125-126) is grammatically not a complete sentence.

Publication may be considered after these mandatory changes.

Author Response

Thank you

Round 2

Reviewer 1 Report

The authors adequately addressed my comments. 

Reviewer 2 Report

The manuscript has been improved, accordingly.